# Using of human capital management in small and medium-sized enterprises in context of Industry 4.0

Nikola Štaffenová[ID]1⊙*, Alžbeta Kucharčíková²⊙, Lukáš Falát²⊙

**1** Department of Managerial Theories, Faculty of Management Science and Informatics, University of Zilina, Zilina, Slovakia, **2** Department of Macro and Microeconomics, Faculty of Management Science and Informatics, University of Zilina, Zilina, Slovakia

⊙ These authors contributed equally to this work.
* nikola.staffenova@fri.uniza.sk

**Citation:** Štaffenová N, Kucharčíková A, Falát L (2025) Using of human capital management in small and medium-sized enterprises in context of Industry 4.0. PLoS One 20(5): e0320568. https://doi.org/10.1371/journal.pone.0320568

## Abstract

Human capital management (HCM) helps manage and develop human capital (HC) in businesses through investments that increase the value of HC and contribute to improving the financial situation of companies. Industry 4.0 (4IR) presents new challenges especially for small and medium-sized enterprises (SMEs). The article aim is to find out how important SMEs perceive investments in HC, whether they evaluate the effectiveness of investments in HC and which of the selected HCM activities related to digitization they consider essential. It was used basic scientific methods, Cronbach's alpha, descriptive statistics, and the chi-square test. It was found that in 2020 up to 49.56% and in 2021 up to 63.94% of respondents consider the funds spent on employees as investments in HC. In connection with digitization, it is striking that up to 76.98% of SMEs in 2020 and 68.34% of respondents in 2021 did not use a personnel information system (HRIS) and an even greater number of companies did not even prepare reports for the HC area. The originality of the article lies in the processing of a two-year questionnaire survey and in the intersection of HCM and Industry 4.0. This is a very current topic, which is also emphasized by the European Union. Future research could focus on employee training models (HC development), or on the culture of innovation and adaptability in companies that digitalize.

## 1. Introduction

The concept of human capital management (HCM) is essential for all types of businesses, not just the large ones. Large businesses have an HCM concept and use digital technologies. However, it was necessary to determine whether the HCM concept and selected digital technologies (established IS for HCM) are also used in Slovak small and medium-sized enterprises (SMEs).

**Data availability statement:** All relevant data are within the paper and its Supporting Information files.

**Funding:** This article was supported by NextGenerationEU through the Recovery and Resilience Plan for Slovakia under the project No.17R05-04-V01-00005. The funders had no role in study design, data collection and analysis, decision to publish, or preparation of the manuscript.

**Competing interests:** The authors have declared that no competing interests exist.

Companies are currently trying to introduce new technologies and use new procedures and recommendations in various management activities [1]. As the environment is currently adapting to the challenges of the 4IR, the basis of which is digitalization, great importance is placed not only on investments in new technologies and IS, i.e., in physical capital but also on investments in HC. These can take three forms, namely investments in education, health, and work ergonomics. According to several authors (Hitka et al., 2021; Forsyth, 2023; Sellar and Zipin, 2019) [2–4], the first of the mentioned forms of investment is the basis of success for the future of businesses. According to Stachová, and Stacho (2017) [5], even employee training should be ensured not only when employees are needed to fulfil company goals but also at the time of their dismissal. Well-informed and educated employees also contribute to the easier implementation of changes in companies [6].

Recently, SMEs have been and are affected by various factors - COVID-19, inflation, energy crisis. SMEs can react more flexibly to changes in the market and make decisions faster since smaller processes need to be changed compared to large enterprises. With the onset of digitalization, the financing of such technologies can be a problem but also the workforce, which is unlikely to have the required knowledge and skills. According to Šafránková et al. (2020) [7], radical technological changes in business models associated with 4IR naturally change workforce requirements and workforce management approaches.

Therefore, in the article, it was focused on the investment of SMEs in Slovakia in HC but also the use of HRIS and subsequent reporting. The article aims to explain the basic terms in the field of HCM and to find out how significant SMEs in Slovakia perceive investments in HC, whether they evaluate the effectiveness of investment in HC and which of the selected HCM activities related to digitalization they consider substantial for their future functioning. It also was investigated whether SMEs, which consider investments in HC as a form of reward, also consider investments in HC as a substantial tool for increasing interest in employment in these SMEs.

The article consists of four chapters the second is an overview of the theoretical knowledge of domestic and foreign authors who researched the concept of HCM, education, and SMEs. The third chapter is the methodology, followed by the results found and, finally the conclusion with a discussion.

## 2. Theoretical backgrounds

### 2.1. Small and medium-sized enterprises (SMEs)

According to the European Commission - Eur-Lex (2003) [8] and the European Commission (2020, 2023) [9,10], there are four basic types of enterprises in Slovakia, namely micro, small, medium, and large businesses. The criterion for SMEs is that they employ less than 250 employees, that their annual turnover does not exceed 50 million euros or that their annual balance sheet does not exceed 43 million euros.

SMEs are the driving force of economies around the world [11–14]. In Slovakia, they account for up to 99.9% of all businesses, while in 2021 up to 59.4% of them were natural persons (self-employed), and 40.6% were legal entities (business

companies). The largest representation of SMEs in Slovakia in 2021 was in the business services sector, followed by construction, trade and industry. The smallest number of SMEs in Slovakia operates in the sector of accommodation and catering services and agriculture. The number of SMEs in Slovakia will increase every year, despite adverse situations, which include the COVID-19 disease that plagued the whole world, high inflation, and the energy crisis [15].

Due to the COVID-19 disease, the employees of most companies were forced to work from home, and this was not the case for the employees of SMEs. However, since SMEs do not have separate IT departments, it is difficult to ensure the operation of the company without disruption by unauthorized persons - hackers. Employees worked with sensitive company data from unsecured environments when they are working from home. Every business in Slovakia in 2021 were exposed to an average of 418 cyber-attacks per week. It was mainly spam, phishing, scanning, or ransomware. These types of cyber threats can cause considerable damage to businesses in the form of obtaining sensitive data or classified documents. Options for protecting SMEs in a virtual environment include network segmentation, regularly updated software, current and functional backups, secure e-mail communication, use of a password generator and manager, use of multi-factor authentication and, finally, employee training in the given area [16].

The World Bank (2023) [17] has been measuring the performance of individual economies since 2005 using the ease of doing business index. The index determines which country has the most suitable environment for establishing and operating a business. The index evaluates 41 indicators within ten topics - setting up a business, obtaining building permits, connecting to electricity, registering with authorities, obtaining funds, protecting minority investors, paying taxes, cross-border trading, enforcing contracts and resolving insolvency. The last available data are from 2020 [18] when Slovakia was ranked 45th with a score of 75.6 (max. 100.0). Slovakia's neighbouring countries were ranked as follows: Austria in 27th place (78.7), Poland in 40th place (score 76.4), Czech Republic in 41st place (score 76.3), Hungary in 52nd place (score 73,4).

The GEM (Global Entrepreneurship Monitor) indicator from 2021 points to the fact that in Slovakia there was low support for creativity or other competencies already in primary and secondary schools (2.89 Slovak score, 3.60 European Union average). Areas in which emphasis is placed on entrepreneurship support at primary and secondary schools (Slovakia score of 2.49, 3.11 European Union average) or government subsidies for new technologies (Slovakia score of 2.65, 4.11) were also rated poorly. European Union average) [19].

## 2.2. Human capital management and education in context of Industry 4.0 and digitalization

HCM reflects talent management, learning and development in a business. This process drives capabilities, leadership, commitment, and performance both individually and organizationally [20]. According to Armstrong (2007) [21], the goal of HCM is to find out the answers to the questions: What are the main drivers of value for the business? What capabilities does the business have? In what way will it acquire, develop, or maintain the given abilities? In what way is it necessary to create and develop a corporate culture and environment? How can it realize that the knowledge created in the company is recorded and at the same time used effectively? Search Financial Applications (2016) [22] understands HCM as an approach to employees that sees them as asset (human capital) whose current value can be measured while its future value can be increased through investments.

A company can invest in HC through three basic types of investments that relate to health, working conditions and education [23].

Given the 4IR and digitalization, investment in HC as a form of education is the most important of all. Increasing the value of HC is also substantial from the point of view that employees can keep up with new technologies. Considering this, it is appropriate and necessary to use training, courses and training processes that are focused on the type of knowledge and skills that are necessary and necessary when working with digital technologies [24,25].

To not only improve the overall production processes in the company but also increase its competitiveness in the market [26], the new revolution requires the introduction of robotization and automation with the ability of intelligence (Bayram,

İnce, 2018). During 4IR, the emphasis should be placed on the training of employees especially on technical skills, because they are more difficult to acquire, and it is also more difficult for business managers to find people with such skills [27]. The importance of education in technological development is confirmed, for example, by Shultz (1961) [28], Kianto et al. (2017) [29], and Carbonaro et al. (2022) [30] who claim that it is people and their knowledge, skills and knowledge that are needed in the implementation of digital technologies in enterprises.

The importance and inevitability of education are also supported by Hricová and Madzinová (2021) [31], who claim that education should firstly help people to remain attractive to employers in the labour market and secondly, that employers should not fire employees at the time of implementing digital technologies, but rather invest in their education. In addition, there is also an increase in the competitiveness and performance of the company [32].

4IR requires highly qualified personnel who will have the necessary IS/ICT and digital knowledge and skills. Only in this way will it represent added value for the company, but also for the country's economy itself [33]. The nature of the work will also change, normal routine activities will be performed by machines and equipment (modern IS/ICT), and therefore employees will have to supplement their education or retrain [34,35]. Digital abilities and skills are considered the most substantial group of people's abilities, skills and knowledge given 4IR. Thanks to them, employees can use digital technologies (IS/ICT) in such a way that they also understand them. It makes work with computers and other IS/ICT more easily. Therefore, according to Lorincová et al. (2018) [36], it is necessary to discuss the issue of individual motivational and educational programs created for individual employees adapted to the needs of employees.

The authors talk about the mutual interaction of employees with modern digital technologies. Some say that the elements of 4IR are necessary if a company wants to improve its processes, increase its competitiveness, and increase work productivity and employee motivation. The introduction of elements such as automation, digitalization, or robotization will cause employees to work much more efficiently than when the company did not use such elements [37].

According to the conducted study, SMEs feel threatened by 4IR and its technologies. In-house knowledge is considered crucial, because if business managers do not continuously train their employees, they will not be able to evaluate the usability, efficiency and, ultimately, even the benefits and meaning of the introduced technologies. Without the necessary knowledge, they are likely to rate technologies as unpromising [38].

The process of digitalization within 4IR makes it possible to use various systems for HCM. For example, the Human Capital Management System is a cloud solution for systems such as Workday, Oracle HCM Cloud, and SAP Success-Factors. It enables basic payroll management functions (e.g., applications for employees, managers, etc.) and support for workforce, talent, or performance management [39].

Another collective of authors claims that 4IR consists of the maximum use of IS/ICT for the development of various production technologies [40–42]. Authors Neumann and Winkelhaus (2021) [43] consider 4IR to be another form of digitalization and integration of information and communication technologies into individual business processes.

During 4IR, a new type of HCM emerged namely digital HCM (electronic human resource management, eHRM). This type of HCM represents the use of information systems and information and communication technologies in the HCM process. The concept is based on internet or intranet technology, which facilitates and accelerates personnel processes. This HCM is often separated from HRIS, but several researchers have confirmed that both concepts perform almost identical tasks. It consists of several parts, such as recruitment, employee selection, performance management, training, compensation and benefit provision, employee self-service systems, HR website or HR portal [44–51].

eHRM is divided into three basic types according to which corporate personnel processes it is used – operational, relational, and transformational [52,53].

The authors' research (Stacho et al., 2023) [54] shows that businesses in the EU, regardless of the sector in which they operate, are aware of the need for digitalization innovations to remain competitive. In their research study, Jankelová and Joniaková (2022) [55] found that the direct relationship between the speed of strategic decision-making and the innovative performance of micro-enterprises and small enterprises did not turn out to be significant. However, information sharing plays a substantial role

   

in this relationship. According to research by Tay et al. (2021) [56], most SMEs try new technologies in practice, mainly expecting an increase in efficiency. He included orientation towards knowledge, technology, capital, education, and workforce among the main challenges for SMEs. The survey results show that SMEs still have limited knowledge about 4IR and are not well prepared for its implementation. The smaller the size of the company, the more obvious the situation. The vast majority do not even realize what opportunities the introduction of digitalization and new technologies offers. However, for this radical change to take place in SMEs, they must understand the entire 4IR that can bring them added value. However, in most enterprises, they do not take any measures regarding new technologies. A knowledge gap has appeared, so it would be better to work on eliminating the deficit of professional knowledge and skills. Since SMEs are financially undersized, the government should help in this regard within the framework of economic policy and various financial schemes.

## 3. Methodology

The article aims to explain the basic terms in the field of HCM and to find out how significant SMEs in Slovakia perceive investments in HC, whether they evaluate the effectiveness of investment in HC and which of the selected HCM activities related to digitalization they consider substantial for their future functioning. It also was investigated whether SMEs, which consider investments in HC as a form of reward, also consider investments in HC as a substantial tool for increasing interest in employment in these SMEs.

With the help of seven research questions, it is wanted to find out whether SMEs in 2020 and in 2021 really apply the following principles and activities within HCM, including their impacts:

**RQ1:** Does the SME consider funds spent on employees as forms of investment in HC?

**RQ 2:** Does the SME evaluate the effectiveness of the investment in HC?

**RQ 3:** Does the SME only evaluate the effectiveness of the investment in education?

**RQ 4:** Does the SME consider investments in HC as a form of reward?

**RQ 5:** Does the SME consider investments in HC as a tool for increasing interest in employment in the SME?

**RQ 6:** Does the SME evaluate HC data using HRIS?

**RQ 7:** Do SME reports include a section on HC?

Based on the research questions, two scientific hypotheses emerged, and it established, which will help to better understand the perception of the HCM concept among SMEs in Slovakia in 2021:

**H1:** If SME considers the funds spent on employees as investments in HC, then the effectiveness of these investments is determined.

**H2:** If investments in HC are taken as a form of reward in the SME, they will be a substantial tool for job seekers to increase interest in employment in the enterprise.

In the article, were used methods such as analysis, synthesis, induction, and deduction, but also methods such as modelling, content analysis, which analysed domestic and foreign professional literature dealing with the issue of HC and HCM. It also used the method of comparison, and the statistical methods, it was calculated Cronbach's alpha for the reliability findings of the questionnaire survey (1st September 2020–31st August 2021). The companies that filled out the online questionnaire gave their informed consent to participate in the anonymous questionnaire survey by clicking the box in the online questionnaire. It also used descriptive statistics and used the chi-square test to detect dependencies between variables.

The questionnaire consisted of two parts, while the first was the identification part. The second part consisted of the core of the questionnaire, which was divided into eight areas. Within the identification part, the option of one answer was used. In the core of the questionnaire, a Likert scale from 1 to 5 was used, while its interpretation depended on the answer, which could refer to the actual application (reality), in which case 1 – Do not apply, 2 – Do not apply yet, but it is considering applying 3 – Cannot assess, 4 – Apply partially, and 5 – Apply, or degrees of importance (future), when 1 – Unimportant, 2 – Rather unimportant, 3 – Cannot assess, 4 – Apply partially, and 5 – Apply completely.

For the evaluation of the questions from the questionnaire, it chooses the criterion by which it determined that it was considered as "Important" and "Apply" those answers where the respondents marked the answer 4 or 5 on the Likert scale. On the contrary, "Unimportant" and "Do not apply" are considered on a Likert scale of answers 1–2. It has listed the answers with a value of 3 separately because the given answers will tell more about how informed managers and business owners are in the case of the concept of HCM and digitalization.

The article processes data from a questionnaire survey, which was carried out in 2020 and 2021. The respondents were managers or owners of Slovak SMEs. It sets a confidence level of 95% and a significance level of α = 5% (0.05). In 2020, it was received responses from 682 respondents, which represents a 3.26% error rate with a preserved confidence interval and a 78% return rate of relevant responses. One year later, our sample set consisted of 477 respondents, which represents a 4.14% error rate and a 92.27% return of relevant responses.

## 4. Results

Using Cronbach's alpha (α) indicator, it was verified the reliability and consistency of the created and analysed questionnaire. The use of this indicator was possible because the entire core of the questionnaire consists of answers to which the respondents answer in the form of a Likert scale. Cronbach's alpha reached a value of 0.98 (α > 0.70). It means that the analysed questionnaire is consistently reliable and can be used to examine the issue of readiness of economic entities for the introduction of digital technologies and their impact on HC.

It was focused on the SME category (Fig 1) which consists of three size groups of enterprises divided by the number of employees - micro (1–9), small (10–49) and medium (50–249) enterprises.

If it founds out in the survey that the company uses the HCM concept, then it focused on investments in HC (Fig 2), which it also used for the needs of article.

In 2020, almost half (49.56%) of respondents and in 2021 up to 63.94% (305) of respondents considered funds spent on education, health, or work ergonomics as forms of investment in HC. Up to 54.69% of subjects in 2020 and up to 62.26% (297) of subjects in 2021 thought that it was important for businesses to perceive these funds for HC investments.

According to Olexová (2018) [57], the evaluation of vocational training focused on specific knowledge and skills should mainly include the benefits and costs of training and the calculation of return on investment. However, in 2020, only 32.55% (222) of respondents devoted themselves to evaluating the effectiveness of investment in HC, while to 39.74% of respondents did not implement this activity in the company. In terms of the level of importance for the future of the company, this activity is considered important by 42.82% of subjects. Also, for that reason, the values for this question were higher in 2021, and 42.98% (205) subjects were devoted to the evaluation of the effectiveness of the investment in HC.

| Size category | 2020 | | 2021 | |
|---|---|---|---|---|
| | Relative abundance | Absolut abundance | Relative abundance | Absolut abundance |
| Micro (1 to 9) | 268 | 39,30 % | 245 | 51,36 % |
| Small (10 to 49) | 212 | 31,08 % | 133 | 27,88 % |
| Medium (50 to 249) | 202 | 29,62 % | 99 | 20,76 % |
| **Sum of SME** | **682** | **100 %** | **477** | **100 %** |

**Fig 1. Number of respondents in terms of size category.**

| Investments in HC | 2020 | | | 2021 | | |
|---|---|---|---|---|---|---|
| | Do not apply (1 and 2) | Cannot assess (3) | Apply (4 a nd5) | Do not apply (1 and 2) | Cannot assess (3) | Apply (4 and 5) |
| Finances spent on education, health, and health and safety are forms of investment in HC | 160 | 184 | 338 | 103 | 69 | 305 |
| The subject evaluates the effectiveness of the investment in HC | 271 | 189 | 222 | 153 | 119 | 205 |
| The subject only evaluates the effectiveness of investment in education | 321 | 186 | 175 | 182 | 132 | 163 |
| Investments in HC are a form of reward | 283 | 218 | 181 | 141 | 111 | 225 |
| Investments in HC are a tool for increasing interest in employment in the entity | 246 | 201 | 235 | 126 | 102 | 249 |

**Fig 2. Investments in HC – a real application.**

The number of companies dealing with the issue should increase in the future, as in 2021, this activity was considered important by 50.10% of respondents.

The next question is related to the previous one it focuses only on evaluating the effectiveness of education. In 2020, only 25.66% of subjects evaluated education, which is an even smaller number than when evaluating the effectiveness of overall investments in HC. It is striking that almost half of the respondents (47.07%; 321) did not evaluate the training provided in the company at all. As many as 36.07% of respondents did not consider the evaluation of education important even in the future. The values increased slightly in 2021 when 34.17% of subjects were involved in the evaluation. However, there are still 65.83% of entities that did not perform this activity. The question in the column devoted to the degree of importance for the future acquires more favourable values. In 2021, up to 43.19% of respondents considered the evaluation of education substantial for their future functioning.

Surprisingly, up to 41.50% of companies in 2020 did not consider investments as a form of reward for employees. From this, it can be concluded that employees were mainly rewarded financially, and non-financial rewards were used minimally in the company (only 26.54%). Within the degree of importance for the future, the subjects were divided into three groups of almost identical size, as 34.46% of subjects did not consider investments in HC as a form of reward, 31.96% of subjects could not decide, and 33.58% of subjects considered it substantial within the future operation of the company. In 2021, when the COVID-19 disease hit Slovakia in a big way it was impossible to carry out almost any leisure activities because the government issued various measures that prohibited routine activities of people and businesses, and people devoted themselves to education. And that's why up to 47.17% of subjects considered education at that time to be a form of reward. Education is also understood positively as part of the future of the company when up to 52.62% of subjects considered education to be substantial.

| Impact of HCM | 2020 | | | 2021 | | |
|---|---|---|---|---|---|---|
| | Do not apply (1 and 2) | Cannot assess (3) | Apply (4 and 5) | Do not apply (1 and 2) | Cannot assess (3) | Apply (4 and 5) |
| Using IS in HCM, HC data is evaluated | 331 | 194 | 157 | 175 | 151 | 151 |
| The subject's reports also contain a section on HC | 329 | 174 | 179 | 186 | 119 | 172 |

**Fig 3. Impacts of HCM on the subject – real application.**

The last question in the section on HC investment asks if SMEs see HC investment as a tool to help them recruit new employees. In 2020, only 34.46% of entities used investments in HC to attract new employees as a form of benefit, while up to 41.79% of entities considered it very important for the future of the company in the same year. In 2021, up to 52.20% of entities used these benefits to recruit employees, and up to 55.77% of entities considered it substantial for the future functioning of the company.

The following two questions address the impact of the HCM concept on the subject (Fig 3). The core of the information system is the data stored in the database [58]. It founds that SMEs in 2020 rarely (up to 48.53% did not use at all, 28.45% probably did not use) did not use personnel IS for evaluating data related to HC. Surprisingly, 38.12% of subjects did not even consider it significant for their future in 2020, and 33.58% of subjects could not express themselves in the case of the importance of personnel IS. A small change occurred in 2021 when 31.66% of subjects used personal IS (36.69% did not use IS at all, and 31.66% did not know if they had an established personal IS). The perception of personnel IS changed towards the future when up to 41.09% of subjects considered it important for the proper functioning of the company.

Businesses in Slovakia make various reports publicly available, such as financial statements, notes to the financial statements, or the annual report. The task of the reports is to inform the company's employees about the implemented activities and results of the company, about its situation within the competition and the prepared necessary measures for improvement, and thus to better motivate and engage all employees in achieving and improving the company's performance. Some documents include a section on HC, so it founds out whether SMEs include the area of HC in their reports or not. In 2020, almost half of respondents (48.24%) said they do not include HC data in their reports. And only 30.50% of subjects think that it is significant for the company to state this. In 2021, 36.06% of entities published HC data, while 38.99% were fundamentally against their publication. Almost half (45.70%) of subjects in 2021 considered it significant for the future of companies that HC data be published and included in company reports.

Fig 4 shows the actual application of seven analysed questions, and Fig 5 captures the degree of importance. The data in both figures are presented in percentages.

Within the actual application in both years (2020 and 2021), the option "Do not apply (1 to 3)" predominates in almost all analysed questions. The exception is the question regarding funds spent on investments in HC – education, ergonomics, and health status. Regarding this question, the respondents in 2020 were divided almost exactly in half - 50.44% do not apply, and 49.56% apply. A year later, even up to 63.94% consider these spent funds as a form of investment in HC. From this, it can be concluded that companies list these expenses as cost items in their accounting but at the same time consider it to increase and develop HC in their company. A positive phenomenon also occurred in 2021 with the question of whether the investments in HC implemented by the company a tool are to increase the interest of the unemployed in

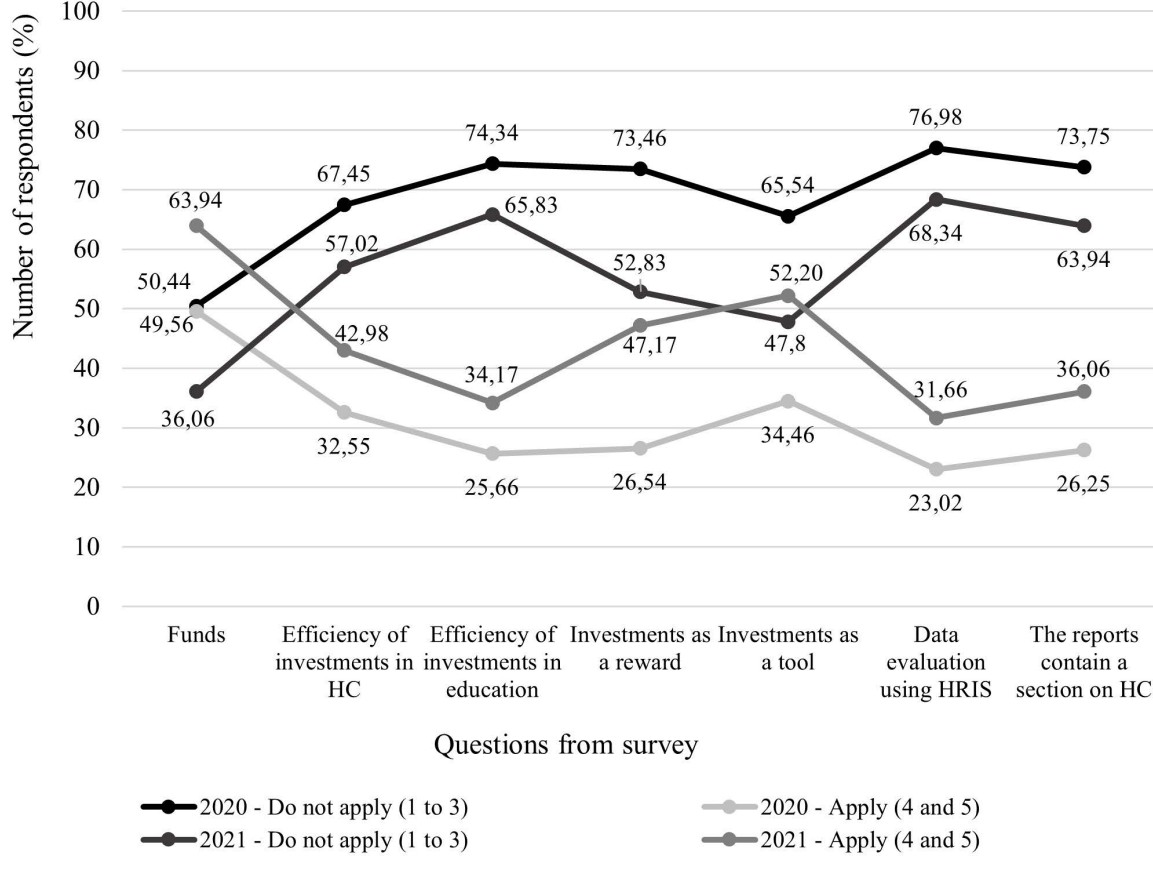

**Fig 4. Comparison of the real application in 2020 and 2021.**

working in the given company. At that time, up to 52.20% of respondents said that they use investments in HC as a tool to attract more job seekers. This situation could also be affected by the disease of COVID-19, when employers discover that part of their business processes can be automated and digitized, and there is no need for the physical participation of employees. So that they don't have to fire their employees, they invested in their education or the ergonomics of work, which changed with the advent of digital technologies.

When analysing the elements of HCM, it was expected that the importance of the elements for the future of the company would exceed the negative answers. However, as is clear (Fig 5), in 2020 this happened only with financial resources when 54.69% of respondents stated that it is substantial for the company to consider the financial resources spent on education, ergonomics and health as a form of investing in HC. In 2021, respondents gave more positive answers, up to 4 questions out of seven. In the first, 54.69% of respondents did so. For the second question, 50.10% of respondents answered that it is important for the company to monitor the effectiveness of investments in HC. This is perhaps because owners and managers want to have an overview of how much they are investing in HC but also what it will bring them and when they can expect a return on their investment. The overwhelming majority (52.62%) of respondent's state that it is important that investments in HC are understood as a form of reward not only by the company but also by the employees in whom it is invested. Investments in HC, which include education, work ergonomics, or health status, can be included by companies among the benefits that the company offers to its current employees and job seekers in the process of recruiting and selecting employees. At the time of the COVID-19 disease, which was still present in Slovakia in 2021, up to 55.77% of business owners and managers said that they considered these investments to be a substantial

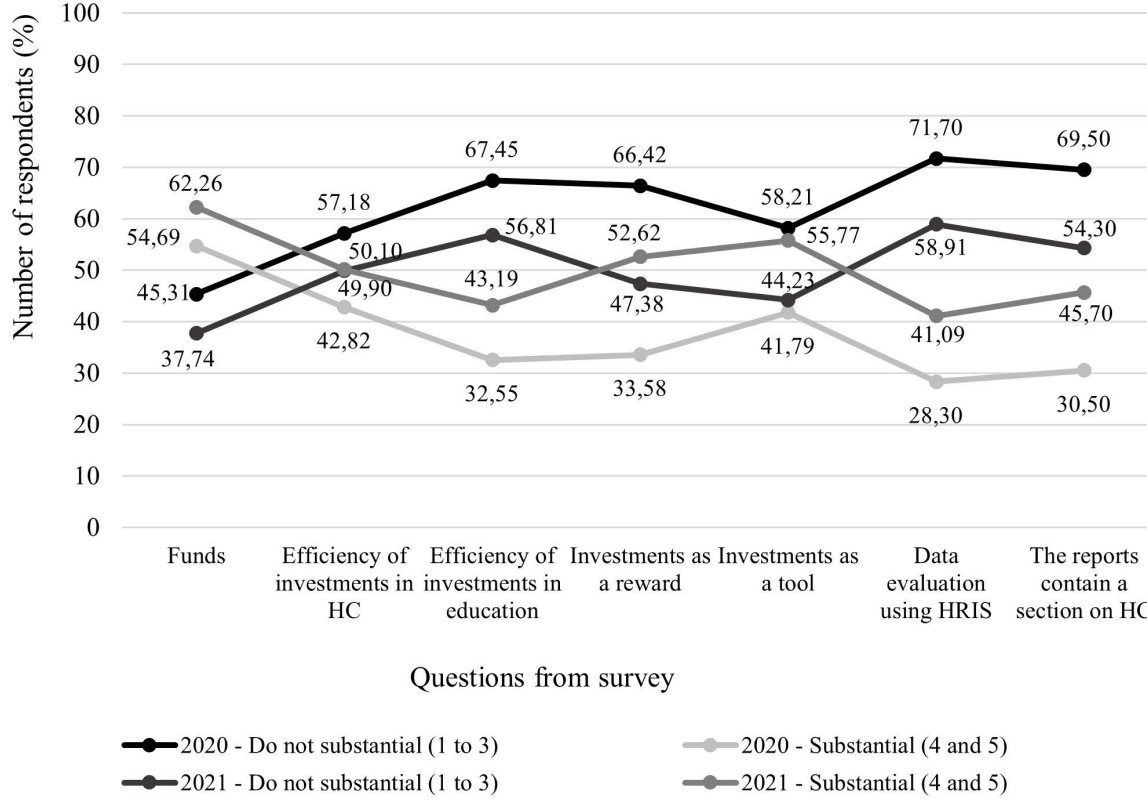

**Fig 5. Comparison of importance level in 2020 and 2021.**

tool for recruiting employees. It is still a striking fact that companies do not perceive the use of HRIS as relevant for their future (substantial for only 41.09%) as well as for the reports to contain a section dedicated to the area of HCM (substantial for only 45.70%). It was analysed the selected seven questions from 2020 through descriptive analysis (Fig 6).

For the mentioned seven parts of HCM, at least 23% of respondents stated that they implemented the given activity, which also corresponds to the value of the modus indicator. At 23%, its value was at the level of 1.00. While the HCM element, where up to 49% of respondents said they apply it, reached a mode value of 4.00. The spiciness values take on negative values in all cases, which means that the data distribution in the file is flatter. Thus, several extreme values occur in the analysed data set. The coefficient of skewness usually takes on positive values, which means that the given HCM elements have a left-skewed distribution of values in a set in which there are lower values. The standard deviation from the arithmetic mean is in the interval from 1.23 to 1.34.

It was choosing the same procedure when analysing the data from 2021 when it was analysed the selected seven questions through descriptive statistics (Fig 7).

When analysing the selected seven questions in 2021, a positive impact of the COVID-19 disease was noted. In the given year, the number of respondents who implemented the given activities increased. This is also evidenced by the average indicator, which in 2021 ranges from 2.80 to 3.52. At the same time, the year before, its values ranged from 2.51 to 3.30. The mode value also increased when for four questions out of seven the answer was at level 4, which can be interpreted that the business is at least partially implementing the activity. The values of the standard deviation, i.e., deviations from the arithmetic mean, range from 1.29 to 1.40. Both the spiciness coefficient and the skewness coefficient acquire exclusively negative values. It can say that this is a flat distribution of data that is skewed to the right, and the values are located further from the mean, while there are more extreme (higher) values.

| Functions of descriptive statistics | Funds | Investments in HC as a tool | Efficiency of investments in HC | Investments in HC as a reward | The reports contain a section on HC | Efficiency of investments in education | Data evaluation using HRIS |
|---|---|---|---|---|---|---|---|
| Number | 682 | 682 | 682 | 682 | 682 | 682 | 682 |
| Mean | 3,30 | 2,90 | 2,80 | 2,70 | 2,58 | 2,57 | 2,51 |
| 95 % confidence level – upper | 3,40 | 3,00 | 2,89 | 2,79 | 2,68 | 2,67 | 2,60 |
| 95 % confidence level – lower | 3,21 | 2,81 | 2,70 | 2,60 | 2,48 | 2,47 | 2,41 |
| Median | 3,00 | 3,00 | 3,00 | 3,00 | 3,00 | 3,00 | 3,00 |
| Mode | 4,00 | 3,00 | 3,00 | 3,00 | 1,00 | 1,00 | 1,00 |
| Std Dev | 1,23 | 1,27 | 1,29 | 1,24 | 1,34 | 1,27 | 1,26 |
| Variance | 1,52 | 1,60 | 1,66 | 1,53 | 1,81 | 1,61 | 1,58 |
| Kurtosis | -0,69 | -1,01 | -1,10 | -0,96 | -1,09 | -1,07 | -1,02 |
| Skewness | -0,45 | -0,06 | 0,01 | 0,10 | 0,30 | 0,21 | 0,27 |
| Range | 4,00 | 4,00 | 4,00 | 4,00 | 4,00 | 4,00 | 4,00 |
| Min | 1 | 1 | 1 | 1 | 1 | 1 | 1 |
| Max | 5 | 5 | 5 | 5 | 5 | 5 | 5 |
| Sum | 2 254,00 | 1 979,00 | 1 908,000 | 1 839,00 | 1 759,00 | 1 753,000 | 1 710,00 |

**Fig 6. Descriptive statistics of selected parts of HCM in 2020.**

For a more thorough examination of the issue, it was decided to establish two hypotheses, which it was statistically verified thanks to the data from the 2021 questionnaire and additional secondary sources (Fig 8).

**H1:** If an economic entity considers the funds spent on education, health, and work ergonomics as investments in HC, then the effectiveness of these investments is determined.

**H2:** If investments in HC are taken as a form of reward in the company, they will be a substantial tool for job seekers to increase interest in employment in the business.

To statistically verify the hypotheses, the Kolmogorov-Smirnov test was calculated, which speaks about the normality of the data distribution in the analysed set, and the Levene test, which tests the agreement of the variances of the data in the set. Both tests came out at 0.000, which is a value lower than the established level of significance (0.05). It follows that hypotheses can be tested only with non-parametric tests - the chi-square test of the independence of two variables.

Since the value of the calculated chi-square test for both hypotheses (H1, H2) were much higher than its tabular value, and the determined level of significance was higher than the calculated p-value, both hypotheses (H1 and H2) were confirmed. For both hypotheses, moderate dependence was also identified when Pearson's C had a value of 0.45 to 0.47.

| Functions of descriptive statistics | Funds | Investments in HC as a tool | Investments in HC as a reward | Efficiency of investments in HC | Efficiency of investments in education | The reports contain a section on HC | Data evaluation using HRIS |
|---|---|---|---|---|---|---|---|
| **Number** | 477 | 477 | 477 | 477 | 477 | 477 | 477 |
| **Mean** | 3,52 | 3,23 | 3,10 | 3,04 | 2,82 | 2,82 | 2,80 |
| **95 % confidence level – upper** | 3,63 | 3,35 | 3,22 | 3,17 | 2,94 | 2,94 | 2,91 |
| **95 % confidence level – lower** | 3,40 | 3,12 | 2,98 | 2,92 | 2,70 | 2,69 | 2,68 |
| **Median** | 3,00 | 3,00 | 3,00 | 3,00 | 3,00 | 3,00 | 3,00 |
| **Mode** | 4,00 | 4,00 | 4,00 | 4,00 | 3,00 | 1,00 | 3,00 |
| **Std Dev** | 1,29 | 1,30 | 1,33 | 1,36 | 1,32 | 1,40 | 1,31 |
| **Variance** | 1,67 | 1,70 | 1,78 | 1,84 | 1,75 | 1,96 | 1,71 |
| **Kurtosis** | -0,50 | -0,88 | -1,06 | -1,14 | -1,16 | -1,29 | -1,10 |
| **Skewness** | -0,77 | -0,51 | -0,38 | -0,24 | -0,03 | -0,01 | -0,03 |
| **Range** | 4,00 | 4,00 | 4,00 | 4,00 | 4,00 | 4,00 | 4,00 |
| **Min** | 1 | 1 | 1 | 1 | 1 | 1 | 1 |
| **Max** | 5 | 5 | 5 | 5 | 5 | 5 | 5 |
| **Sum** | 1 678,00 | 1 543,00 | 1 477,00 | 1 452,00 | 1 346,00 | 1 344,00 | 1 334,00 |

**Fig 7. Descriptive statistics of selected parts of HCM in 2021.**

It means that H1 is valid: If an economic entity considers the financial resources spent on education, health, and work ergonomics as investments in HC, it also determines the effectiveness of these investments. H2 is also valid: If investments in HC are taken as a form of reward in the company, for job seekers, according to SMEs, they will be a substantial tool for increasing interest in employment in SMEs.

## 5. Discussion and conclusions

SMEs are of great importance for every economy because they cover a high industrial share and, therefore the integration of 4IR into SMEs, as well as other possible forms of support, for example from the government [59], is substantial not only for the development of these enterprises but also every economy. The sustainability of SMEs in a globalized economy depends on several factors, e.g., also from the use of the collaborative economy, i.e., the sharing of employees between organizations and countries. The impact of 4IR on employee sharing was addressed by Bednarz et al. (2023) [60], who in their research outlined the importance of human capital, including its components, which are knowledge and skills. During

| Signpost | H1 | H2 |
|---|---|---|
| Degrees of freedom (df) | 4 | 4 |
| Chi square test ($\chi^2$) | 118,65 | 138,36 |
| Chi square tabl. | 9,49 | 9,49 |
| P-value | 0,0000 | 0,0000 |
| Significance level ($\alpha$) | 0,05 | 0,05 |
| The result | **confirmed** | **confirmed** |
| Addiction (Pearson´s C) | 0,45 | 0,47 |
| | medium | medium |

**Fig 8. Statistical hypothesis testing.**

the 4IR, IT skills and knowledge prove to be the most important [61]. According to available studies, effective customer relationship management is one of the factors positively affecting the competitiveness of SMEs [62]. The willingness of SMEs to invest in the HC of their employees also contributes to a better reputation with customers, thus SMEs show that they value their employees and consider them an asset of the company.

Several authors emphasize [63,64] the lack of financial resources needed to introduce the elements of 4IR (for example, IS for HCM) in SMEs, and at the same time, they are aware of the need to invest in the education of their employees.

In the evaluation of H1, it was demonstrated that the financial resources spent on education, work ergonomics and the health status of employees are considered investments in HC in SMEs in Slovakia. If this is how they perceive investments in HC, then they also evaluate them. It is a positive finding because in the future it can assume a higher interest of SMEs in investing in HC.

It was surprising that in 2020 up to 41.50% of SMEs did not consider investments as a form of reward for employees. However, in 2021, apparently also due to the COVID-19 pandemic, the situation changed. As many as 47.17% of SMEs changed their view of education and considered it a form of reward. In the case of these companies, the evaluation of H2 confirmed that if investments in HC are taken as a form of reward in the company, they will also be considered by the company as a substantial tool for increasing interest in employment in the company for job seekers.

According to Vetráková et al. (2018) [65], thanks to the use of information technology, the hiring of new employees is easier to apply in practice because companies can present themselves and obtain information about potential job applicants. It is also important for companies to determine the way to effectively set motivational factors to retain talent in the company [66].

In Slovakia, in the past, many SMEs justified their employees' lack of interest in investing in HC by the lack of financial resources. In 2020, only 49.56% of managers in SMEs perceived funds spent on education, health, and safety at work as investments in HC. It is gratifying that, apparently also because of the COVID pandemic, businesses began to value their employees more, and in 2021 up to 63.94% of managers began to perceive these funds as investments. Unfortunately, still few entities evaluate these investments, which also applies to investments in education. If companies were to implement these activities as part of HCM, it would lead to better targeting of investments and improvement of the financial situation, which is also confirmed by Olexová (2018) [57].

The conclusion of the study (Polat, Karakuş, 2019) [67] showed that SMEs in Turkey, which contribute to the development of the country, are not sufficiently informed about the given issue. The biggest prejudices about 4IE refer to the high

investment need and the insufficient qualification of the workforce. It deduces that in Slovakia too, the insufficient use of IS for HCM, as one of the elements of 4IR, is caused, in addition to the lack of financial resources, by insufficient information and insufficient knowledge about the benefits of increasing the efficiency of working with people and, ultimately, increasing the efficiency of the entire company. Could the introduction of IS bring them?

It is striking that SMEs in Slovakia do not use HRIS in 2020 and 2021 (76.98% in 2020, 68.34% in 2021) and do not even consider it important for their future (71.70% in 2020, 58.91% in 2021). Procuring an HRIS can be financially demanding for SMEs, or such companies can have the personnel and payroll agenda developed by external companies that specialize in this.

A lot of information about the need to implement HRIS and the need for education in connection with digitalization can also be obtained by the employees by the fact that SMEs will regularly prepare and share reports with the employees, which will also include a report on HC. It is interesting that SMEs do not include a section on HCM in their reports and do not consider it important. The reason for this is the fact that small businesses in Slovakia must only prepare financial statements if they account in the double-entry bookkeeping system. This report does not include any sections where companies can describe the concept of HCM. An annual report in which companies could describe the HCM concept must be compiled only by those companies that meet at least two of the three criteria – number of employees over 30, assets over 2 million euros, net turnover over 4 million euros. Even if SMEs decide to outsource personnel activities, they must pay attention to employee education. It is because, in the era of 4IR, businesses may be exposed to various technologies that they must be able to control. An example of this was the introduction of electronic communication between legal entities and state institutions in Slovakia, which included the tax office, social insurance, health insurance, and the labour office [68].

The importance of HCM lies mainly in keeping capable employees in the company, whose activity is reflected in its overall functioning. It is therefore, in the interest of companies to strive to ensure the protection of current employees in the future, to adapt to increasingly demanding conditions and to increase the qualifications of employees. According to Safránková, and Sikýr (2020) [69], technological progress associated with 4IR naturally affects all businesses, including SMEs, which play a substantial role in the economic and social development of society. SMEs should therefore apply appropriate strategies, policies, and procedures in the field of human resource management to attract enough talented and motivated people and ensure the development of their potential.

Professions that are expected to be among the most important are, for example, technicians, experts in the field of information and technology [70,71], digital human resources, digital marketing experts, data analysts and Big Data management experts, or internal audit experts [72]. In the future, it is therefore expected that, on the contrary, activities that machines are unable to perform correctly and efficiently will be performed by humans. These are mainly activities that require the ability to manage people, communication skills, cognitive skills, logical reasoning, high creativity but social and emotional skills [73].

The ability to work with data and make decisions based on data will represent one of the most substantial tasks in the jobs of the future [74]. Specialized skills, which are also extremely necessary for the future of the labour market, were also tracked in detail. These are mainly skills related to the field of product and digital marketing and, of course, human-computer interaction [75].

A study by Lertpiromsuk et al. (2021) [76] for the food industry focused on SMEs and found that graduating students from these fields are not sufficiently prepared for practice. They lack basic social skills, and personal prerequisites, as well as technical and methodical skills that are important for 4IR. Although these gaps in experience decrease after starting a job, there is still a shortage in both types. Therefore, it is necessary to not be left without additional education even while the student is already working so that the industry can grow, and the skills gap is minimized.

The rapid growth of modern technologies leads to a shortage of personnel with specialized skills. Even today, employers are unable to find workers with the right qualifications, and globally it is estimated that this leads to losses of more than 1.6 trillion dollars [77].

For this reason, it will be necessary for managers in SMEs to change their perspective on the funds spent on increasing the HC value of their employees. These finances should be understood as investments in HC and not as increased costs for the company. At the same time, they must realize and convey to people the idea that investments in HC are understood by the company at the same time as a form of reward and a motivational factor.

According to Müller et al. (2018) [78], companies have a better chance of overcoming financial constraints when trying to implement 4IR if they have an active strategy and the ability to acquire knowledge, and know-how and create networks.

Based on the above facts, it can be concluded that although initial investments in employee education and retraining are high, they are ultimately lower than when companies start to face innovation differences, and their competitiveness will be lower and lower. In addition to efforts to retain and train current employees, it will be necessary to attract, retain and integrate new talent in businesses, which is emphasized by Lizbetinova et al. (2020b) [66]. Human resource managers should be prepared to redefine the necessary HC and recruitment. It is time for companies to realize that their willingness to invest in the HC of their employees can act as motivation [79] and attract new, capable, and trained applicants.

With the introduction of 4IR, employees began to be replaced by machines. Despite the advanced age, there are gaps in human-centred management that need to be addressed. It is therefore necessary that 4IR investments are closely integrated with the needs of the human factor, which can bring a more motivated and satisfied workforce and a workforce with increasing abilities and skills, as individual components of HC [43]. Another challenge is the organizational challenge, which is associated with solving problems with an ageing workforce, and implementing smart procedures and processes, but also includes how to deal with the situation of employee benefits after adopting new technology [80]. It will also be necessary for managers in companies to focus on new methods of education, using the new 4IR technologies, for example, artificial intelligence, and machine learning [81].

At the end of the discussion, it is important to summarize several practical consequences that flow for SMEs within the framework of the digitalization process. First, it is about increasing productivity and efficiency because digitalization allows to streamline business processes, including their management. Furthermore, SMEs gain better access to markets not only on a national but also international scale, because they can engage in online sales and digital forms of marketing campaigns, which can reach a larger range of people. SMEs that introduce digital technologies into their processes increase their competitiveness in the markets. Thanks to faster and more flexible reactions to market changes or monitoring trends, SMEs can provide better services and products to their customers. This is also related to the improved customer experience because digital technologies enable a personalized approach not only to employees, but also to customers. Thanks to digitalization, SMEs have better access to obtaining funds, because their credibility in the eyes of investors or financial institutions increases. Finally, it is about developing business models that will enable SMEs to transition to subscription-based models or to provide products and services via digital platforms. Finally, digitalization in SMEs can also increase employment in IT sectors, which will lead to the development of expertise and an improved workforce on the labour market.

There are several avenues for future research that the article could explore. This could include employee development (HC development) in the form of education, examining approaches to training so that the company can fully utilize digital technologies, or specific models of employee education. Furthermore, future research could focus on the culture of innovation and adaptability in companies that are digitizing, examining how companies can support employees' willingness to accept change and technological innovations. Current research could continue by analyzing the role of managers in SMEs in digitization, i.e., how managers support the company's innovation environment and how they increase productivity within teams through digital technologies. It is also possible to examine the cooperation of SMEs with educational institutions to secure highly qualified personnel necessary for the digitization process.

The limitation of the research is mainly its implementation in the conditions of one Central European country (Slovak Republic). It is also possible to consider the fact that the survey was conducted in 2020–2021 (i.e., the research is 3 years ago) as a limitation. However, despite the above limitations, the research is of high quality and the methodology examines enough responses from respondents, which is also confirmed by the statistical results presented in the manuscript.

## Author contributions

**Conceptualization:** Nikola Štaffenová, Alžbeta Kucharčíková.

**Data curation:** Nikola Štaffenová, Lukáš Falát.

**Formal analysis:** Nikola Štaffenová.

**Funding acquisition:** Alžbeta Kucharčíková.

**Methodology:** Nikola Štaffenová, Lukáš Falát.

**Project administration:** Alžbeta Kucharčíková.

**Resources:** Nikola Štaffenová.

**Software:** Nikola Štaffenová, Lukáš Falát.

**Supervision:** Alžbeta Kucharčíková.

**Validation:** Nikola Štaffenová, Alžbeta Kucharčíková.

**Visualization:** Nikola Štaffenová, Lukáš Falát.

**Writing – original draft:** Nikola Štaffenová, Alžbeta Kucharčíková, Lukáš Falát.

**Writing – review & editing:** Nikola Štaffenová.

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
