## [Decision Letter · Decision Letter 0]

21 Jan 2025

PONE-D-24-39079Using of Human Capital Management in Small and Medium-Sized Enterprises in context of Industry 4.0PLOS ONE

Dear Dr. Štaffenová,

Thank you for submitting your manuscript to PLOS ONE. After careful consideration, we feel that it has merit but does not fully meet PLOS ONE’s publication criteria as it currently stands. Therefore, we invite you to submit a revised version of the manuscript that addresses the points raised during the review process.

**ACADEMIC EDITOR: **Please take into account the reviewers recommendations and change your paper accordingly.

We look forward to receiving your revised manuscript.

Kind regards,

Valentina Diana Rusu, PhD

Academic Editor

PLOS ONE

2. You indicated that ethical approval was not necessary for your study. We understand that the framework for ethical oversight requirements for studies of this type may differ depending on the setting and we would appreciate some further clarification regarding your research. Could you please provide further details on why your study is exempt from the need for approval and confirmation from your institutional review board or research ethics committee (e.g., in the form of a letter or email correspondence) that ethics review was not necessary for this study? Please include a copy of the correspondence as an ""Other"" file.

“This article was supported by project Scientific Grant Agency of the Ministry of Education of the Slovak Republic VEGA 1/0273/22, by project Slovak Research and Development Agency APVV-20-0004 The Effect of an Increase in the Anthropometric Measurement of the Slovak Population on the Functional Properties of Furniture and the Business Processes.”

5. Please remove your figures from within your manuscript file, leaving only the individual TIFF/EPS image files, uploaded separately. These will be automatically included in the reviewers’ PDF.

Reviewers' comments:

Reviewer's Responses to Questions

**Comments to the Author**

1. Is the manuscript technically sound, and do the data support the conclusions?

Reviewer #1: Yes

Reviewer #2: Yes

2. Has the statistical analysis been performed appropriately and rigorously? 

Reviewer #1: Yes

Reviewer #2: Yes

3. Have the authors made all data underlying the findings in their manuscript fully available?

Reviewer #1: Yes

Reviewer #2: Yes

4. Is the manuscript presented in an intelligible fashion and written in standard English?

Reviewer #1: Yes

Reviewer #2: Yes

5. Review Comments to the Author

Reviewer #1: The manuscript is interesting source of information about human capital management (HCM) in the context of challenges posed by Industry 4.0, particularly for small and medium-sized enterprises (SMEs). The authors explore the significance of investments in human capital and assess the effectiveness of these efforts, with a focus on digitization as a key driver of contemporary business transformation.

The two-year study, (2020, 2021), provides valuable insights. It highlights a growing awareness among SMEs of viewing employee-related expenses as investments in human capital. However, the findings also reveal significant gaps in the adoption of HR information systems (HRIS) and the preparation of reports on human capital management, which could hinder SMEs' ability to effectively manage their workforce amid digital transformation.

A strength of the study lies in its use of diverse research methods, including descriptive statistics, chi-square tests, and Cronbach's alpha. Furthermore, the article successfully integrates HCM with the technological challenges of Industry 4.0, contributing valuable insights to this area of research.

General remarks:

- The title of the paper is well-formulated and accurately reflects its content.

- The abstract is well-written – it includes the aim and scope of the research, highlights the applied research methods and analyses, and emphasizes the most important findings obtained.

- The scientific background is appropriately developed. The theoretical introduction is based on well-selected academic publications.

- In scientific papers, it is recommended to avoid the use of personal forms such as "me", "I", "we", etc. - that must be changed in entire manuscript.

- Both the research method and the statistical analysis are correctly and clearly described - Cronbach's alpha reached a value of 0.98 (α > 0.70). It means that the analysed questionnaire is consistently reliable and can be used to examine the issue of readiness of economic entities for the introduction of digital technologies and their impact on HC.

- The obtained results are appropriately discussed in the text, which contributes to the article being highly rated in terms of scientific quality.

While the article offers numerous merits, it could benefit from a more detailed discussion of practical implications, particularly in terms of supporting SMEs in their digitization efforts. Nevertheless, this work represents a significant contribution to understanding human capital management in the era of digital transformation and will be of interest to both researchers and practitioners in the field.

Reviewer #2: Dear Authors,

Thank you for your manuscript, whose aim is to find out how important small and medium-sized enterprises perceive investments in human capital, whether they evaluate the effectiveness of investments in human capital, and which of the selected human capital management activities related to digitization they consider essential. This paper addresses an important topic and has several strengths. In my opinion, to be published, your paper minor improvements. Please see my comments.

1. The abstract should provide not only an overview of the work; main findings; main conclusions or interpretations; but recommendations for future research.

2. Final remarks should provide a future research agenda. You outline specific avenues for future research or enhancements.

3. The authors should provide limitations of the research. The Limitations section should include separate limitations that occurred in data collection, data analysis, conclusions, and generalizability.

4. In conclusion, the results of the research presented in the paper are valuable and should be presented to the scientific audience. However, to be published, your paper minor improvement.

6. PLOS authors have the option to publish the peer review history of their article (what does this mean? ). If published, this will include your full peer review and any attached files.

**Do you want your identity to be public for this peer review?** For information about this choice, including consent withdrawal, please see our Privacy Policy .

Reviewer #1: No

Reviewer #2: No

---

## [Author Response · Author response to Decision Letter 0]

8 Feb 2025

We have included responses to the comments of reviewer 1, reviewer 2, and the editor in the attached files, which are labeled Respond to Reviewer/Editor.

---

## [Decision Letter · Decision Letter 1]

21 Feb 2025

Using of human capital management in small and medium-sized enterprises in context of Industry 4.0

PONE-D-24-39079R1

Dear Dr. Štaffenová,

We’re pleased to inform you that your manuscript has been judged scientifically suitable for publication and will be formally accepted for publication once it meets all outstanding technical requirements.

Kind regards,

Valentina Diana Rusu, PhD

Academic Editor

PLOS ONE

Additional Editor Comments (optional):

Reviewers' comments:

Reviewer's Responses to Questions

**Comments to the Author**

1. If the authors have adequately addressed your comments raised in a previous round of review and you feel that this manuscript is now acceptable for publication, you may indicate that here to bypass the “Comments to the Author” section, enter your conflict of interest statement in the “Confidential to Editor” section, and submit your "Accept" recommendation.

Reviewer #1: All comments have been addressed

Reviewer #2: All comments have been addressed

2. Is the manuscript technically sound, and do the data support the conclusions?

Reviewer #1: Yes

Reviewer #2: Yes

3. Has the statistical analysis been performed appropriately and rigorously? 

Reviewer #1: Yes

Reviewer #2: Yes

4. Have the authors made all data underlying the findings in their manuscript fully available?

Reviewer #1: Yes

Reviewer #2: Yes

5. Is the manuscript presented in an intelligible fashion and written in standard English?

Reviewer #1: Yes

Reviewer #2: Yes

6. Review Comments to the Author

Reviewer #1: I accept the article as it stands. The authors have made all the suggested corrections. I am recommending an article for publication.

Reviewer #2: Thank you for your manuscript, whose aim is to find out how important small and medium-sized enterprises perceive investments in human capital, whether they evaluate the effectiveness of investments in human capital, and which of the selected human capital management activities related to digitization they consider essential.

The revised version incorporates my suggested comments to a sufficient extent. In my opinion, the paper can be published in its present form.

7. PLOS authors have the option to publish the peer review history of their article (what does this mean? ). If published, this will include your full peer review and any attached files.

**Do you want your identity to be public for this peer review?** For information about this choice, including consent withdrawal, please see our Privacy Policy .

Reviewer #1: No

Reviewer #2: No

---

## [Editor Report · Acceptance letter]

PONE-D-24-39079R1

PLOS ONE

Dear Dr. Štaffenová,

I'm pleased to inform you that your manuscript has been deemed suitable for publication in PLOS ONE. Congratulations! Your manuscript is now being handed over to our production team.

Kind regards,

on behalf of

Dr. Valentina Diana Rusu

Academic Editor

PLOS ONE